# Biogenesis, Functions, Interactions, and Resources of Non-Coding RNAs in Plants

**DOI:** 10.3390/ijms23073695

**Published:** 2022-03-28

**Authors:** Haoyu Chao, Yueming Hu, Liang Zhao, Saige Xin, Qingyang Ni, Peijing Zhang, Ming Chen

**Affiliations:** Department of Bioinformatics, College of Life Sciences, Zhejiang University, Hangzhou 310058, China; haoyuchao@zju.edu.cn (H.C.); huym@zju.edu.cn (Y.H.); zhaoliang97@zju.edu.cn (L.Z.); 22107033@zju.edu.cn (S.X.); 22107035@zju.edu.cn (Q.N.)

**Keywords:** ncRNA, plant, ncRNA resource, ncRNA function, ncRNA interaction

## Abstract

Plant transcriptomes encompass a large number of functional non-coding RNAs (ncRNAs), only some of which have protein-coding capacity. Since their initial discovery, ncRNAs have been classified into two broad categories based on their biogenesis and mechanisms of action, housekeeping ncRNAs and regulatory ncRNAs. With advances in RNA sequencing technology and computational methods, bioinformatics resources continue to emerge and update rapidly, including workflow for in silico ncRNA analysis, up-to-date platforms, databases, and tools dedicated to ncRNA identification and functional annotation. In this review, we aim to describe the biogenesis, biological functions, and interactions with DNA, RNA, protein, and microorganism of five major regulatory ncRNAs (miRNA, siRNA, tsRNA, circRNA, lncRNA) in plants. Then, we systematically summarize tools for analysis and prediction of plant ncRNAs, as well as databases. Furthermore, we discuss the silico analysis process of these ncRNAs and present a protocol for step-by-step computational analysis of ncRNAs. In general, this review will help researchers better understand the world of ncRNAs at multiple levels.

## 1. Introduction

In the last few years, a number of non-coding RNAs (ncRNAs) have been described in plants involved in several processes, ranging from RNA maturation, splicing, regulation of transcription, post-transcriptional RNA modifications, and nucleosome remodeling. Therefore, it is unquestionable that ncRNAs play a significant role in gene regulatory network [1,2,3,4]. With extensive transcriptome analysis, up to 90% of the eukaryotic genome is transcribed into RNA, of which only 1–2% corresponds to protein-coding mRNA [5,6]. Although the remaining transcripts lack minimal protein-coding capacity and poorly conserved sequences [2,5,7], the emergence of ncRNAs as novel ribose regulators of gene expression sheds light on the so-called “dark matter” of the genome.

Current studies have revealed that ncRNAs can be transcribed from DNA sequences in protein-coding genes, intergenic or intronic regions [8]. In terms of their regulatory roles, ncRNAs could be divided into two major categories in plants (Figure 1). Among them, housekeeping ncRNAs are necessary for fundamental biological processes of life, so the content is relatively constant. Regulatory ncRNAs vary in size, shape, and accumulation patterns, so their expression is temporal and spatially specific [9]. Notably, any ncRNAs classification system is defined as an intelligent construct that is unlikely to perfectly reflect nature [10]. To date, many ncRNAs have not been described in plants, such as PIWI-interacting RNA (piRNA), an animal-specific small silencing RNA [11,12], enhancer RNAs (eRNAs), which play critical role in transcriptional activation in mammalian cells and are transcribed from enhancers [13,14,15], and Y RNAs, which are necessary for DNA replication in humans [16,17].

So far, the biogenesis of some regulatory ncRNAs has been clearly described [18,19,20]; however, siRNA and tsRNA are poorly defined in plants. As further studies have shown, ncRNAs participate in the maintenance of homeostasis in plants by ncRNA-associated interaction with other biomolecules and microorganisms, which is of great significance to growth, development, differentiation, and reproduction of plants.

With the advancement of high-throughput RNA-seq technologies, the diversity of ncRNAs world has been unveiled. To date, numerous studies have applied RNA-seq technology to discover known and novel classes of ncRNAs in diverse tissues and developmental stages [21]. These precious data have been mined and stored in public databases.

In this review, we mainly focus on five regulatory ncRNAs, including microRNA (miRNA), small interfering RNA (siRNA), tRNA-derived small RNA (tsRNA), circular RNA (circRNA), and long ncRNA (lncRNA) in plants. Here, we will show a regulator ncRNAs biogenesis landscape and give a hypothesis of tsRNAs biogenesis in plants based on previous studies. Besides, we will discuss the ncRNAs interaction with DNA, RNA, protein, and microorganism for helping our understanding of the dynamic inter-molecular networks within plant cells. Furthermore, we will summarize the algorithms, databases, and RNA-seq-based analysis pipelines of the regulatory ncRNAs in the field of plants to incentivize researchers to make greater use of RNA-seq technologies and bioinformatics approaches to discover the individual, as well diverse, regulatory ncRNAs in the exceedingly complicated landscape of ncRNAs.

## 2. Biogenesis and Functions of ncRNAs in Plants

Currently, massive endogenous ncRNAs with various regulatory potentials have been discovered in various plant species [22,23]. Based on their average size, regulatory ncRNAs could be further categorized into small RNAs (18–30 nt), medium-sized ncRNAs (31–200 nt), and lncRNAs (>200 nt). In addition, it can be classified into linear or circular according to its morphology (Figure 1). In general, 200 nt is regarded as the dividing line in the regulatory ncRNAs world, but this size consideration is arbitrary because circRNAs, eRNAs, and promoter-associated transcripts (PATs) have displayed variable lengths [8]. Recently, regulatory small RNAs (sRNAs), namely, miRNA and siRNA, are considered to have tiny sizes but play important roles in response to stress or environmental changes by regulating the expression of target genes [24,25,26,27]. Likewise, lncRNAs were contemplated as transcriptional noise but later gained importance as one of the wide-ranging and heterogeneous groups of ncRNAs [28]. Notably, unlike other linear regulatory ncRNAs, circRNAs are a novel class of ncRNAs that lack free 5′ and 3′ terminus, which have been extensively explored in the past few years [29]. Besides, many small ncRNAs derived from tRNAs, called tsRNAs, have also been identified in plants with a broad size range of 15–42 nt [30,31,32]. Here, we classified tsRNAs as regulatory ncRNA according to their diverse functions (Figure 1). In general, the functions of some regulatory ncRNAs are similar, while a few are distinct, nevertheless overlapping in silencing pathways [33]. Next, we will introduce their biogenesis and functions in plants in detail.

### 2.1. miRNA

miRNA biogenesis is a multistep process involving transcription, processing, modification, and assembly of the RNA-induced silencing complex (RISC) (Figure 2A) [34,35,36]. First, primary miRNAs (pri-miRNAs) are transcribed by RNA POLYMERASE II (Pol II) containing hairpin RNA secondary structures. Then, an RNase III family DICER-LIKE (DCL) enzyme, usually DCL1 [37], assisted by HYPONASTIC LEAVES 1 (HYL1) and SERRATE (SE), cleaves from the base of the pri-miRNA hairpin to yield a precursor-miRNA (pre-miRNA) hairpin and cleaves again to release a miRNA/miRNA* duplex [38]. Next, the 3′-most nucleotides of the initial miRNA/miRNA* duplex are then 2′-O-methylated by the nuclear HUA ENHANCER 1 (HEN1) protein for stabilizing miRNA [39]. Finally, most mature miRNA strands are incorporated into ARGONAUTE 1 (AGO1) in the nucleus (unlike in animals, where it occurs in the cytoplasm [40]), with the removal of the miRNA* strand and the transport of the miRNA-AGO1 complex to the cytoplasm, where miRNAs induce post-transcriptional gene silencing by transcript cleavage and translation repression [24,35,41,42].

As post-transcriptional gene regulators, miRNAs are up- or down-regulated for improving plant productivity and stress tolerance in numerous species [43,44]. Therefore, studying the expression patterns of miRNAs can help us better understand the regulatory networks of stress response and environmental adaptation. In general, miRNAs have several features in regulatory pathways. (1) Evolutionarily conserved RNAs tend to have conserved targets in related plant species. For example, miR159 targets *MYB* genes and is down-regulated in salt-stress responses among *Arabidopsis* [45], tobacco [46], and kidney bean [47]. miR172 targets *AP2* genes for regulating floral development in *Arabidopsis*, rice, soybean, barley, and maize [27]. A list of conserved miRNAs suggests a common regulatory mechanism across different species. (2) A single miRNA can participate in a variety of stress responses and developmental processes. For instance, cold-inducible miR393 targets *TIR1/AFB* genes and is up-regulated for enhancing cold tolerance [48]. miR393 is also induced by PAMP flagellin (flg22) to down-regulate the levels of *TIR1/AFB* genes for antibacterial defense [49]. Besides, miR393 is also involved in regulating arbuscule formation [50], inhibiting root elongation, and promoting lateral root initiation [51]. (3) Multiple miRNAs can participate in one biological process. For example, in rice, miR156, miR396, and miR397 cooperate in the regulation of grain size. miR156, miR393, and miR444 participate in tillering together [27]. (4) The expression patterns of miRNAs rely on the specific condition. As mentioned above, miRNAs are up- or down-regulated according to specific stress or specific tissue. Another example can also be supported. miR1425 will influence the number of fertile pollen grains by regulating a pentatricopeptide repeat (PPR)-containing protein under cold stress [52].

### 2.2. siRNA

According to the mode of action, siRNA can be simply divided into three secondary categories, namely *trans*-acting siRNAs (ta-siRNA), heterochromatic siRNAs (hc-siRNA), and natural antisense siRNAs (nat-siRNAs). However, in fact, ta-siRNAs belong to the so-called “secondary siRNAs” category, including ta-siRNA and phased siRNAs. Here, since many of the known ta-siRNAs are also phased [10], we use ta-siRNA instead of “secondary siRNAs” for discussion. Generally, they both play a role in transcriptional gene silencing by complementary target mRNAs or directing DNA and histone methylation through RNA-directed DNA methylation (RdDM) process [53,54].

ta-siRNAs are generated from *TAS* genes (Figure 2B) [24,55]. Firstly, *TAS* genes are transcribed into single-stranded RNAs by RNA Pol II, and then they loose the cap and mostly also the poly-A end upon miRNA-AGO1 complex guided cleavage. Secondly, the 5′ or 3′ cleavage fragments are protected by SUPPRESSOR OF GENE SILENCING 3 (SGS3) and converted to double-stranded RNA (dsRNA) by RNA-dependent RNA polymerases 6 (RDR6) [56]. Finally, they are methylated and processed into 21–24 nt ta-siRNAs by HEN1 and various DCL activities. The 21–22 nt size class are loaded onto AGO1 or AGO7 to induce post-transcriptional gene silencing of complementary target mRNAs in the cytoplasm, while some ta-siRNAs are incorporated into AGO4/6 to guide RNA Pol V-mediated de novo DNA methylation of *TAS* genes [54]. In *Arabidopsis*, miR173 targets *TAS1* and *TAS2* genes to generate ta-siRNAs [55,57]. The TAS1 ta-siRNAs target the heat stress transcription factor genes, *HEAT-INDUCED TAS1 TARGET 1* (*HTT1*) and *HTT2*, to regulate plant thermotolerance [58].

nat-siRNAs can be divided into two categories, *cis*-NAT-siRNAs and *trans*-NAT-siRNAs. However, only *cis*-NAT-siRNAs have been described in plants. *trans*-NAT-siRNAs remain only a hypothetical possibility. Therefore, in this review, *cis*-NAT-siRNAs are collectively referred to as nat-siRNAs. Previously, nat-siRNAs were thought to be generated by the hybridization of separately transcribed complementary RNAs. However, to date, many of the nat-siRNAs investigated depend on RDR for their accumulation [10,59,60,61,62]. This RDR dependency suggests that the precursor dsRNA did not derive from the hybridization of two separately transcribed, complementary mRNAs. Thus, the biogenesis of nat-siRNAs is not well defined and appears to be very complex with some important unanswered questions. Based on available data, Zhang et al. speculated that there are at least five possible mechanisms to generate nat-siRNAs [63]. However, it is clear that nat-siRNAs can be induced by salt [59], pathogen [63], and control sperm function during double fertilization in *Arabidopsis* [61].

The biogenesis of hc-siRNA begins with the transcription of RNA Pol IV from the intergenic or repetitive genomic regions to generate single-stranded siRNA precursors [64,65,66], which are converted into dsRNA and processed into 24 nt siRNA duplexes. Methylated hc-siRNAs are loaded into AGO4 in the cytoplasm and are transported to the nucleus [67], followed by the recruitment of these hc-siRNA-AGO4 complexes to RNA Pol V transcripts. The subsequent recruitment of DOMAINS REARRANGED METHYLASE 2 (DRM2) catalyzes DNA methylation at RdDM target loci [53,67].

### 2.3. tsRNA

With a broad size range of 15–42 nt, tsRNAs are a new category of regulatory ncRNAs, which are classified into five categories according to the cleavage sites, namely tRF-1s, tRF-2s, tRF-3s, tRF-5s, and tiRNA (Figure 2C). However, the study in plants has just started, and many questions remain to be answered. For example, the biogenesis pathway of tsRNAs in plants is still unclear, and the physiological function of certain tsRNA in plants is currently very limited [68]. In this review, we propose a hypothesis of tsRNAs biogenesis in plants based on previous studies. First, RNA poly III transcribes tRNA gene as precursor tRNA (pre-tRNA) [69], which includes a 5′ leader, a mature tRNA backbone, a 3′ U trailer, and sometimes an intron [70]. Then, the 5′ leader, 3′ U trailer, and intronic sequences are cleaved by RNase P, RNase Z, and tRNA-splicing endonucleases (TSEN) to produce mature tRNA and tRF-1s (tRF-1s could be derived from 3′-end of pre-tRNA) [71,72,73]. The mature tRNA (73–90 nt) forms a secondary cloverleaf structure with a D-loop (left), a T-loop (right), anticodon loop (bottom), a variable loop, and an acceptor stem (Figure 2C). Finally, the mature tRNAs could be cleavaged by Arabidopsis S-like Ribonuclease 1 (RNS1) and/or DCL1 to form tRF-2s, tRF-3s, tRF-5s, and tiRNA (Figure 2C) [30,31]. In mammals, tsRNAs incorporate into silencing AGO and trigger RNA interference [74]. Likewise, AGO-associated tsRNAs have been predicted in *Arabidopsis* and rice [30,75]. An in vitro assay has shown that certain tsRNAs regulate gene expression by translation inhibition, and tsRNA-AGO1 complex tends to target transposable element transcripts and probably maintains genome stability [31,76,77].

### 2.4. circRNA

CircRNAs were first discovered in plant viruses by Sanger’s group in 1976. Studies have shown that circRNAs are circular, single-stranded, and covalently closed RNA biomolecules [78]. The composition of circRNAs can be divided into three categories (Figure 2D). (1) Exonic circRNAs are formed by lariat-driven circularization and intron pairing-driven circularization [79]. (2) Intronic circRNAs are the source of introns generated by the partial degradation of introns after the formation of the lasso structure. (3) Exonic-intronic circRNAs, which are composed of exons and introns, are cyclized during splicing. In 2013, Jeck et al., proposed that exon skipping and intron pairing reduced the distance between splicing sites and promoted the reverse splicing of pre-mRNA [80]. This leads to the deletion of the 3′ and 5′ ends of circRNAs [81]. Several distinct functional mechanisms for animal circRNAs have been identified, suggesting that plant circRNAs may exhibit similar conserved functions. These include miRNA decoys [82], transcriptional modulation [83], translation of circRNAs into small peptides [84]. Besides, circRNAs can play an important role in plant development and stress responses. For example, Vv-circATS1 responds to cold stress by regulating the expression of stimulus-responsive genes in grape [85]. Under dehydration-stressed conditions, many differentially expressed circRNAs have been detected in wheat [86], pear [87], maize, and *Arabidopsis* [88]. These studies suggest that circRNAs have post-transcriptional roles. However, the mechanism of this remains to be elucidated.

### 2.5. lncRNA

The biogenesis of lncRNAs can be divided into five categories according to the transcribed site by Pol II: (1) sense lncRNAs are transcribed on the same strand as exons; (2) antisense lncRNAs are transcribed on the opposite strand of exons; (3) intronic lncRNAs are transcribed on introns; (4) intergenic lncRNAs are located between two distinct genes; (5) enhancer lncRNAs emerge from an enhancer region of protein-coding genes (Figure 2E) [89]. They can control target regulation by multiple ways, including chromatin remodeling [90,91,92], transcriptional repression, RNA splicing and transcriptional enhancer [93,94]. In addition, lncRNAs may encode small peptides (Figure 2E), which are required for various cellular processes [95]. Notably, numerous plant lncRNAs are regulated by abiotic stresses. For example, many differentially expressed lncRNAs have been identified in *Arabidopsis* under drought, cold, salinity, heat, and abscisic acid stresses [96]. Besides, biotic stress-responsive lncRNAs have also been identified in wheat [97], *Arabidopsis* [98], and tomato [99].

## 3. ncRNA-Associated Interaction

Recent studies have demonstrated that ncRNAs are involved in the maintenance of plant homeostasis through ncRNA-associated interaction with DNA, RNA, protein, and microorganism, respectively, and have important implications for plant growth, development, differentiation, and reproduction [100,101]. Typically, these ncRNAs interact with genes or gene products (such as proteins and various RNAs) in the nucleus and cytoplasm region, thereby affecting biological processes and altering their cell fate [102,103,104,105,106,107,108]. Therefore, further discussions of how ncRNAs interact with other biological macromolecules will help advance our understanding of the landscape of the dynamic inter-molecular networks within plant cells.

### 3.1. ncRNAs Interact with RNAs

A series of experimental methods, such as PAR-CLIP [109], HITS-CLIP [110], CLASH [111], and LIGR-seq [112], were developed to define ncRNAs function and how they interact with other RNAs. Notably, LIGR-seq is a novel technology that can be used to detect RNA duplexes at scale without prior knowledge. Besides, several computational studies have also predicted that snoRNAs can interact with other RNA types, thereby regulating biological functions and cell signaling pathways [113].

Furthermore, the interactions among congener and isotypic ncRNAs can influence several biological processes, including epigenetic modifications and translation. For example, miRNA response elements (MRE), such as circRNAs, lncRNAs, and eRNAs, act as competing endogenous RNAs (ceRNAs) with rich implications for gene regulation in various physiological and pathophysiological processes at the post-transcriptional level [114]. ncRNAs can also act as gene sponges to regulate gene expression. It is well known that circRNAs and lncRNAs can generate internal regulatory networks through circRNA/lncRNA–miRNA-gene [115]. To date, the mechanisms of ncRNAs–RNAs interaction have been well defined in mammals and *Homo sapiens* [116,117,118], but little has been studied in plants. Hence, it is emergent to calculate the interactions between ncRNAs and RNAs in plants.

### 3.2. ncRNAs Intact with DNAs

So far, numerous ncRNAs have been reported to be involved in the regulation of gene expression in the nucleus as direct regulators [119,120,121,122]. They may play roles in nucleosome positioning, chromatin marking, and transcriptional regulation. Currently, those ncRNAs that interact with the global genome can be detected through deep-sequencing technology, such as GRID-seq [123].

Furthermore, some lncRNAs have been demonstrated to be able to divide the nuclear region into distinct compartments and participate in the organization of multi-chromosomal regions [124,125,126]. These lncRNAs have a close affinity to chromosomes through nuclear matrix factors, and they also provide favorable advantage for lncRNAs to interact with functional DNA elements related to transcriptional regulation. Therefore, lncRNAs can not only interfere with the expression of protein-coding genes, which are close to lncRNA genes, but also spread throughout the nucleus, close to spatial affinity sites, and regulate the expression of genes on chromosomes [127,128].

### 3.3. ncRNAs Interact with Proteins

ncRNA–protein interactions play a crucial role in regulating cell metabolism. It is widely known that numerous RNA binding proteins (RBPs) can change the fate or function of the bound RNAs in the nucleus region. To date, the RNA–protein regulatory relationship can be detected by CLIP-seq [129]. Therefore, a series of approaches were also developed to reveal how and where ncRNAs interacted with corresponding proteins [110,130].

Generally, ncRNAs achieve and regulate various functions by forming various ribonucleoprotein (RNP) complexes with proteins. For example, snRNPs, which are composed of snRNAs and proteins, can direct both canonical splicing and alternative splicing. Besides, numerous nucleotides in pre-rRNA, pre-snRNA, and pre-tRNA undergo post-transcriptional modification by nucleolar RNP particles [131]. Small ncRNAs, such as miRNA and siRNA, can also influence the regulation of gene expression by interacting with AGO family proteins during RNA interference pathway [132,133,134,135,136].

As for circRNA, research has shown that they can serve as protein sponges to transport proteins into specific subcellular compartments. On the other hand, lncRNA can achieve their function via recruitment, inhibition, and acting indirectly through genome organization and transcription [137]. *ALTERNATIVE SPLICING COMPETITOR* (*ASCO*) lncRNAs have been reported to be regulators of alternative splicing in *Arabidopsis* through interactions with splicing factors [138]. Apparently, with more and more thorough studies of the interactions between ncRNAs and proteins, people may gradually recognize a far more complex regulation network of interactions in plants.

### 3.4. ncRNAs Interact with Microbe

Plants have an animal-like innate immune system [139,140]. When attacked by pathogenic microorganisms, plants recognize pathogen-associated molecules through plasma membrane-associated pattern recognition receptors (PRR) and trigger immunity (PTI) [141]. Some microorganisms inhibit the signal transduction of plant PTIs by secreting small ncRNAs as effector molecules, leading to the occurrence of diseases. In addition, plants also mainly utilize extracellular vesicles to transport small ncRNAs into pathogens to suppress virulence-related genes [60].

Currently known microorganisms regulate their pathogenic capacity based on two modes of small ncRNA regulation. (1) The first way is to regulate their own toxicity through sRNAs derived from microorganisms, such as the sRNA of the entomopathogenic fungi *Metarhizium anisopliae* and *Sclerotinia sclerotiorum*, respectively, in conidia formation, and sclerotia are differentially expressed [142,143]. (2) The second way is to inhibit the small ncRNAs and RNAi pathways of plants through microbial effector proteins to achieve pathogenic effects, e.g., two effectors from the oomycete plant pathogen *Phytophthora sojae* suppress RNA silencing in plants by inhibiting the biogenesis of small ncRNAs [144].

In addition to the innate immune system, plants have evolved two ways of regulating small ncRNAs in response to infection by pathogenic microorganisms. The first way is that plant endogenous sRNAs are involved in the regulation of immune responses. For example, *Arabidopsis* miR863-3p fine tunes plant immune responses during infection by sequentially silencing negative and positive regulators of plant immunity [145]. Another study reported that the disease resistance protein (R protein) SNC1 represses the transcription of miRNA and ta-siRNA loci, probably through the transcriptional corepressor TPR1. This study revealed an additional layer of regulation—a regulatory circuit formed by miRNAs, ta-siRNAs, and NLR proteins to modulate and fine tune the trade-off between plant growth and defense [146]. The second pathway is plant growth through RNAi mechanisms to regulate the infectivity of microorganisms. We know that small ncRNAs are produced by DCLs and act through AGOs to silence target genes [147,148,149]. AGO2 is the only *Arabidopsis* AGO that is highly induced by bacterial infection [150]. In *rhizobia-legume* symbiosis, AGO7 of *Lotus japonicas* is required for TAS3 ta-siRNA biogenesis, and it is important for the development of nitrogen-fixing nodules in plant roots [151,152].

## 4. Bioinformatics Resources for ncRNA Analysis

In the past two decades, substantial research efforts have been devoted to discovering non-coding regulatory RNAs and studying their functions, including miRNAs, lncRNAs, and circRNAs. With the development of next-generation sequencing (NGS), high-throughput sequencing has been widely used to characterize the ncRNA transcriptomes under various conditions, which provides an unprecedented opportunity to discover ncRNAs and identify differentially expressed ncRNA transcripts. However, rapidly growing sequencing data have created challenges for the identification, annotation, and storage of ncRNAs. Here, we systematically summarize the prediction tools, database resources, and integration workflows of various ncRNAs in plants.

### 4.1. ncRNA Prediction Tools in Plants

Next-generation sequencing offers unprecedented opportunities to discover and quantify various ncRNAs. To date, numerous computational methods have been developed for single ncRNA category, such as miRDeep-P2, miR-island, phasiRNAClassifier, NATpare, tsRFinder, RNAplonc, FEELnc, Circle-Map, and CircMarker [153,154,155,156,157,158,159,160,161]. Besides, integrated analysis of multiple ncRNAs has also been published, such as mirTools 2.0 and sRNAtools [154,162]. To select suitable tools and platforms for ncRNA prediction, we systematically summarize the tools and platforms for plant ncRNA prediction and analysis (Table 1).

### 4.2. ncRNA Databases in Plants

With the decreasing cost of NGS sequencing and the advent of various tools for prediction and characterization of ncRNAs, the number of annotated ncRNAs has grown exponentially. Therefore, relevant databases of ncRNAs are emerging rapidly. There are lots of ncRNA-related databases in plants, most of which focus on a single-type ncRNA, such as PmiREN2.0 [212], MepmiRDB [212,213], GreeNC v2.0 [214,215], and AtCircDB v2.0 [216]. Due to the lack of a unified way to access ncRNA information, fragmented data make it challenging and incompatible for ncRNA search and comparison. Here, we comprehensively summarize high-quality databases of various ncRNAs in plants. Data sources and stored ncRNA information are listed in Table 2.

### 4.3. RNA-Seq-Based Pipeline for ncRNA Analysis in Plants

Currently, the research on various ncRNA functions using RNA-Seq and sRNA-seq technology has achieved great success. However, only a few tools and platforms are available for the analysis and prediction of most ncRNAs in plants, such as mirTools 2.0 and sRNAtools [154,162]. Most tools only predict and interpret a single ncRNA, such as miRDeep-*p*, FEELnc, Circle-Map, CPAT, and many other pipelines [159,160,172,245]. Choosing appropriate tools for comprehensive analysis of plant ncRNAs is a great challenge. To overcome this challenge, we summarize a comprehensive workflow for ncRNA analysis (Figure 3), which provides a general analysis pipeline from traditional RNA-seq/sRNA-seq to specific ncRNA identification.

In this workflow, we present a protocol for step-by-step computational analysis of ncRNAs, which mainly consists of three parts: (1) raw fastq data preprocessing, including read quality filtering and 3′ adapter trimming; (2) alignment and annotation; (3) sequence feature analysis, including ncRNA data alignment and novel ncRNA prediction.

## 5. Discussion

Although identification and characterization of biogenesis and fundamental functions of regulator ncRNAs are necessary, efforts to understand the depth and diversity of ncRNAs are the way forward. In the past decade, with the advancement of high-throughput sequencing technology, many bioinformatics methods and tools have emerged, which has enabled the discovery and research of many ncRNAs. Studies have found that ncRNAs mainly include housekeeping ncRNAs, such as tRNA, rRNA, etc., which are necessary to maintain plant life activities, and regulatory ncRNAs, such as miRNA, lncRNA, circRNA, etc., which play a role in regulating the special life process of plants. The interactions between these ncRNAs and the regulation of the expression of other encoded genes constitute a complex regulatory network of ncRNAs in plants.

Compared with numerous studies, such as miRNA and lncRNA, the functional characterization of circRNA is still in its infancy. Even in humans and animals, there are only a few reports on the functional study of circRNAs. Although many plants circRNAs have been identified from plant circRNA databases or sequencing analyses, the functional mechanism of only one circRNA from *Arabidopsis* has been revealed [83]. Due to the overlapping sequence characteristics of circRNAs, it is difficult to knock out circRNAs by traditional RNA interference methods, and the CRISPR-Cas system will enable new strategies for further circRNA function research.

At present, the research of these ncRNAs is more focused on their regulatory mechanisms during the growth and development of plant organs or tissues and in special environments. However, the expression and changes of ncRNAs in different cell types remain unclear [246]. In recent years, only a few reports have investigated the regulatory mechanisms of ncRNAs at the single-cell level in humans or animals [247,248,249], such as Luo et al. using scRNA-seq in three cancer-type data, where a total of 154 characteristic lncRNA genes related to effector, depletion, and regulatory T cell states were identified [249]. With the maturity of scRNA-seq technology, it is believed that the regulatory mechanisms of plant ncRNAs at the cellular level can be analyzed.

## 6. Conclusions

In this review, we systematically summarized the functions and regulatory relationships of major classes of ncRNAs in plants. Besides, we reviewed and summarized the computational methods, tools, and knowledge bases of ncRNA in plants in detail. Notably, we proposed a general protocol for step-by-step computational analysis of different types of ncRNAs to help researchers choose appropriate ncRNA analysis tools and platforms. We hope this work contributes to a better understanding of the complex ncRNA world.

## Figures and Tables

**Figure 1 ijms-23-03695-f001:**
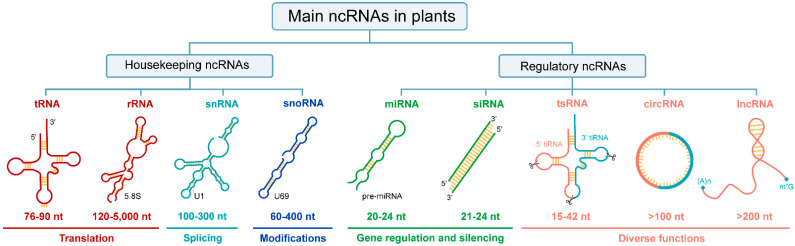
ncRNAs category in plants. From top to bottom, there are primary classification, secondary classification, abbreviations, secondary structures, size, and functions of ncRNAs. Since some ncRNAs contain multiple types, one is selected and annotated with text in the lower right corner of the secondary structure. The sizes of ncRNAs are approximate. Diverse functions include gene expression regulation, translation inhibition, plant immunity, stress response, etc. Abbreviations: tRNA, transfer RNA; rRNA, ribosomal RNA; snRNA, small nuclear RNA; snoRNA, small nucleolar RNA; miRNA, micro RNA; siRNA, small interfering RNA; tsRNA, tRNA-derived small RNA; circRNA, circular RNA; lncRNA, long non-coding RNA; tiRNA, stress-induced tRNA or tRNA halves; nt, nucleotides.

**Figure 2 ijms-23-03695-f002:**
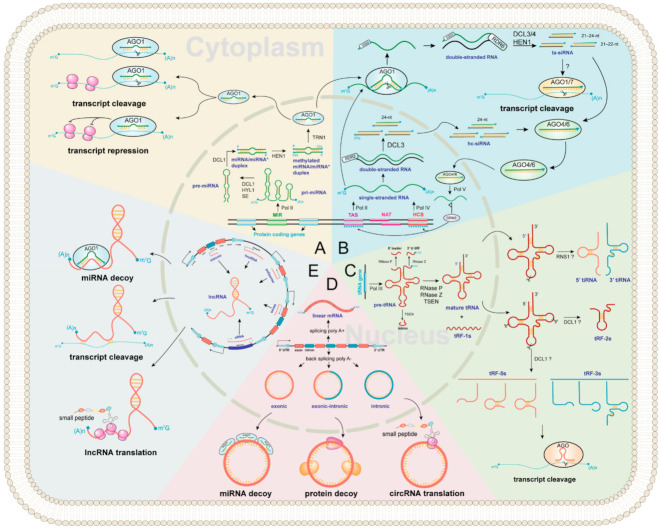
Regulator ncRNAs biogenesis landscape and functions in plants. The inside of the circle represents the nucleus, and the outside represents cytoplasm. All ncRNAs types are marked in purple. The background of each color represents biogenesis and functions of (**A**) miRNA; (**B**) siRNA; (**C**) tsRNA; (**D**) circRNA; (**E**) lncRNA, respectively.

**Figure 3 ijms-23-03695-f003:**
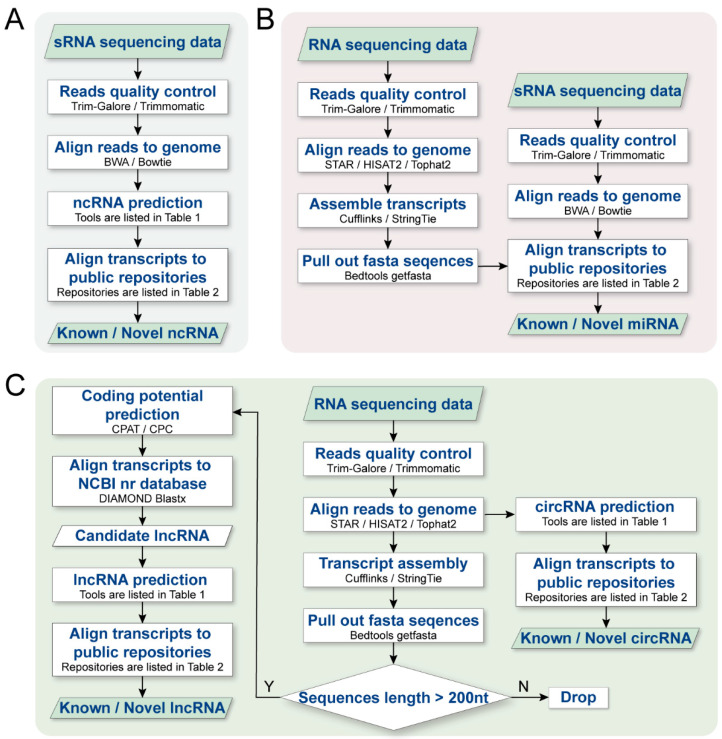
The analysis workflow for differentiating between different classes of ncRNAs in sRNA-seq/RNA-seq datasets. (**A**) General analysis workflow from sRNA-seq data to ncRNA prediction (including data preprocessing, sequence alignment, ncRNA prediction, and related database alignment), (**B**) represents the general analysis process from RNA-seq to lncRNA/circRNA prediction (the lncRNA analysis process includes data preprocessing, sequence alignment, transcript assembly, sequence length filtering, transcript encoding potential estimation, database alignment, and lncRNA prediction. The circRNA analysis process includes data preprocessing, sequence alignment, circRNA prediction, and related database alignment), (**C**) represents the general process from sRNA-seq/RNA-seq to miRNA prediction (where analysis process from RNA-seq to miRNA prediction is like the general analysis process in A, while sRNA-seq to miRNA prediction includes data preprocessing, sequence alignment, miRNA prediction, and related database alignment).

**Table 1 ijms-23-03695-t001:** List of ncRNA prediction tools in plants.

ncRNA Types	Software Package	Platform	Last Update	Link	Ref
miRNA	miRDeep-P2 v1.1.5	Linux	2021.09	https://sourceforge.net/projects/mirdp2/	[153]
	UEA sRNA Workbench	All	2020.05	https://github.com/sRNAworkbenchuea/UEA_sRNA_Workbench	[163]
	MITP v1.1	All	2019.05	https://github.com/wushyer/MITP	[164]
	PmiRDiscVali	Linux	2018.12	https://github.com/unincrna/pmirdv	[165]
	miRPlant v6	All	2018.12	https://sourceforge.net/projects/mirplant/	[166]
	Chimirac v1.5	Web	2018.10	http://wwwdev.ebi.ac.uk/enright-dev/chimira/index.php	[167]
	CAP-miRSeq	Linux	2018.08	http://bioinformaticstools.mayo.edu/research/cap-mirseq/	[168]
	Mirnovod	All	2018.06	https://github.com/dvitsios/mirnovo	[169]
	microRPM	Linux	2018.05	http://microRPM.itps.ncku.edu.tw	[170]
	miRCat2 v4.5	All	2018.05	http://srna-workbench.cmp.uea.ac.uk/mircat2	[171]
	miRDeep-*p* v1.3	Linux	2011.06	https://sourceforge.net/projects/mirdp/	[172]
	sRNAnalyzer	All	2017.12	http://srnanalyzer.systemsbiology.net	[173]
	miRDis	Web	2017.01	http://sbbi.unl.edu/miRDis/index.php	[174]
	miRA v1.2.0	Linux/Mac/	2016.05	https://github.com/mhuttner/miRA	[175]
	miRNA Digger	Windows	2016.01	http://bis.zju.edu.cn/miRNA_Digger/	[176]
	Mir-PREFeR v0.24	All	2015.06	https://github.com/hangelwen/miR-PREFeR	[177]
	mirBayes	All	2015.05	https://github.com/smdouglass/mirBayes	[178]
	Mirinho	Mac	2015.06	http://mirinho.gforge.inria.fr	[179]
	MIRPIPE v1.2	Linux	2014.10	https://github.com/loosolab/mirpipe	[180]
	BioVLAB-MMIA-NGSc	Web	2014.09	http://epigenomics.snu.ac.kr/biovlab_mmia_ngs/	[181]
	MTide v1.0	Linux	2014.09	http://bis.zju.edu.cn/MTide/	[182]
	miRSeqNovel	All	2014.07	https://sourceforge.net/projects/mirseq/?source=navbar	[183]
	miReader	Linux	2014.02	https://sourceforge.net/projects/mireader/	[184]
	eRNA v1.01	Linux	2014.07	https://sourceforge.net/projects/erna/files/?source=navbar	[185]
	plantDARIOc	Web	2013.11	http://plantdario.bioinf.uni-leipzig.de/index.py	[186]
	isomiRIDa	All	2013.10	http://www.ufrgs.br/RNAi/isomiRID/	[187]
	isomiRexca	Web	2013.08	http://bioinfo1.uni-plovdiv.bg/isomiRex/	[188]
	MIReNA v2.0	Linux	2013.08	http://www.lcqb.upmc.fr/mirena/index.html	[189]
	miRAutoa	Linux	2013.04	http://nature.snu.ac.kr/002535831182/software/miRAuto.htm	[190]
	miRPlexa	All	2013.08	https://www.uea.ac.uk/computing/mirplex	[191]
	PIPmiR v1.1-5	All	2012.11	https://bioconda.github.io/recipes/pipmir/README.html	[192]
	mirDeepFindera	Linux	2012.11	http://www.leonxie.com/DeepFinder.php	[193]
phasiRNA	findPhasiRNAs	Linux/Mac	2019.03	https://github.com/Wiselab2/findPhasiRNAs	[194]
	phasiRNAClassifier v1	Linux	2018.11	https://github.com/pupatel/phasiRNAClassifier	[155]
	PhaseTank v1.0	Linux	2014.11	http://phasetank.sourceforge.net/	[195]
ta-siRNA	NATpare	All	2020.05	https://github.com/sRNAworkbenchuea/UEA_sRNA_Workbench	[156]
	NASTI-seq v1.0	Linux/Windows	2017.02	https://ohlerlab.mdc-berlin.de/software/NASTIseq_104	[196]
	NATpipe	Linux	2015.11	www.bioinfolab.cn/NATpipe/NATpipe.zip	[197]
	pssRNAMinerca	Web	2008.05	http://bioinfo3.noble.org/pssRNAMiner/	[198]
tsRNA	tsRFinder v1.0.0	Linux/Mac	2019.05	https://github.com/wangqinhu/tsRFinder	[157]
lncRNA	RNAplonc V1.1	Linux	2021.08	https://github.com/TatianneNegri/RNAplonc	[158]
	PlncRNA-HDeep	Linux/Windows	2021.05	https://github.com/kangzhai/PlncRNA-HDeep	[199]
	CREMA	Linux	2021.06	https://github.com/gbgolding/crema	[200]
	PlncPRO v1.2.2	Linux	2020.05	http://ccbb.jnu.ac.in/plncpro/	[201]
	CNITa	Linux	2019.05	http://cnit.noncode.org/CNIT/download	[202]
	Evolinc I v1.7.5	Linux	2019.02	https://github.com/Evolinc/Evolinc-I	[203]
	PLIT	Linux	2018.09	https://github.com/deshpan4/PLIT	[204]
	lncRNA-screen v.02	Linux	2017.04	https://github.com/NYU-BFX/lncRNA-screen	[205]
circRNA	Circle-Map v1.1.4	All	2021.03	https://github.com/iprada/Circle-Map	[160]
	Rcirc	Linux/Mac	2020.09	https://github.com/PSSUN/Rcirc	[206]
	CircMarker	Linux	2020.07	https://github.com/lxwgcool/CircMarker	[161]
	Ularcirc	All	2020.07	https://github.com/VCCRI/Ularcirc	[207]
	CirComPara v1.1.1	Linux	2020.06	https://github.com/egaffo/CirComPara	[208]
	CIRCfinder	Linux	2019.12	https://github.com/YangLab/CIRCfinder	[209]
	PcircRNA_finder	Linux	2017.11	https://github.com/bioinplant/PcircRNA_finder	[210]
	Acfs	Linux	2017.02	https://github.com/arthuryxt/acfs	[211]
ncRNA	mirTools v2.0	Web	2013.05	http://www.wzgenomics.cn/mr2_dev	[162]
	sRNAtools	Web	2019.12	https://bioinformatics.caf.ac.cn/sRNAtools	[154]

**Table 2 ijms-23-03695-t002:** List of ncRNA repositories and ncRNA interaction repositories in plants.

Database Name	Stored ncRNAs or ncRNA-Associated Interaction	Number of Plant Species	Year	Link	Ref
PmiREN2.0	38,186 miRNA loci and 141,327 predicted miRNA-target pairs	179 plant species	2020	http://www.pmiren.com	[212]
MepmiRDB	9218 miRNAs	29 medicinal plant species	2019	http://mepmirdb.cn/mepmirdb/index.html	[213]
Plant IsomiR Atlas	98,374 templated and non-templated isomiRs from 6167 miRNA precursors	23 plant species	2019	http://www.mcr.org.in/isomir	[217]
Diff isomiRs	33,874 isomiRs	16 plant species	2019	http://www.mcr.org.in/diffisomirs	[218]
miRbase v22	8615 hairpin precursors and 10,414 mature miRNAs	82 plant species	2019	http://www.mirbase.org/	[219,220,221,222,223]
mirEX v2.0	461 miRNAs	3 plant species	2015	http://www.combio.pl/mirex2	[224,225]
miRNEST v2.0	39,122 miRNAs	199 plant species	2014	http://rhesus.amu.edu.pl/mirnest/copy/	[226,227]
CSRDB	10,000 smRNAs	2 plant species	2007	http://sundarlab.ucdavis.edu/smrnas	[228]
sRNAanno	24,630 miRNAs, 22,721 phasiRNA, 22,404,950 hc-siRNAs loci annotations	143 plant species	2021	http://www.plantsrnas.org/	[229]
Small RNA plant genes	2,786,895 sRNAs loci annotations	48 plant species	2020	https://plantsmallrnagenes.science.psu.edu	[230]
PNRD	15,041 miRNAs, 189 ta-siRNAs, 5,573 lncRNAs	150 plant species	2015	http://structuralbiology.cau.edu.cn/PNRD/	[231]
tasiRNAdb	583 ta-siRNAs regulatory pathways	18 plant species	2014	http://bioinfo.jit.edu.cn/tasiRNADatabase/	[232]
PtRFdb	5607 tRFs	10 plant species	2018	http://www.nipgr.ac.in/PtRFdb/	[233]
tRex	1,409,566 tRFs	*Arabidopsis thaliana*	2018	http://combio.pl/trex	[234]
GreeNC v2.0	>495,000 lncRNAs.	78 plant species	2022	http://greenc.sequentiabiotech.com/wiki2/Main_Page	[214,215]
PLncDB	1,246,372 lncRNAs	80 plant species	2021	http://plncdb.tobaccodb.org/	[235,236]
CANTATAdb v2.0	239,631 lncRNAs	39 plant species	2019	http://cantata.amu.edu.pl/	[237,238]
DsTRD	27,687 lncRNAs	*Salvia miltiorrhiza*	2016	http://bi.sky.zstu.edu.cn/DsTRD/home.php	[239]
PLNlncRbase	1187 lncRNAs	43 plant species	2015	http://bioinformatics.ahau.edu.cn/PLNlncRbase	[240]
AtCircDB v2.0	84,685 circRNAs.	*Arabidopsis thaliana*	2019	http://deepbiology.cn/circRNA/	[216]
CircFunBase	1158 circRNAs	7 plant species	2019	http://bis.zju.edu.cn/CircFunBase	[241]
CropCircDB	38,785 circRNAs in maize and 63,048 circRNAs in rice	Rice and maize	2019	http://deepbiology.cn/crop/	[242]
PlantcircBase v6.0	142,115 circRNAs and 68,193 circRNA loci	20 plant species	2018	http://ibi.zju.edu.cn/plantcircbase/	[243,244]

## Data Availability

Not applicable.

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
