# Peer review of "Biogenesis, Functions, Interactions, and Resources of Non-Coding RNAs in Plants"

_ijms, 2022, doi:10.3390/ijms23073695_

Round 1

Reviewer 1 Report

This review focused on the regulatory ncRNAs widely studied in plants, and summarize the roles of these ncRNAs in plant development and response to stress. In addition, they summarized the bioinformatics analysis processes such as ncRNA prediction and annotation, interaction, and important database resources and tools in the field of plant. The manuscript is well structured and well discussed. However, some points should be checked and corrected before its acceptance in this journal. 

Therefore, I recommended the publications of the paper after major revision according to given my comments.

  • The abstract is not clear. Please add the aim and objective of the MS.
  • The study's background should be clearly stated. Describe the introduction and review of the work.
  • In Conclusion, the authors should add the significance of this research, and its potential practical application. Conclusion should be separate.
  • The MS English needs to be improved. The article's English must be carefully checked for grammatical errors.

Author Response

Response to Reviewer 1 Comments

This review focused on the regulatory ncRNAs widely studied in plants, and summarize the roles of these ncRNAs in plant development and response to stress. In addition, they summarized the bioinformatics analysis processes such as ncRNA prediction and annotation, interaction, and important database resources and tools in the field of plant. The manuscript is well structured and well discussed. However, some points should be checked and corrected before its acceptance in this journal. Therefore, I recommended the publications of the paper after major revision according to given my comments.

The abstract is not clear. Please add the aim and objective of the MS.

Response

Thank you for your suggestion. We have revised the abstract, and added aim and objective at line 18-24.

The study's background should be clearly stated. Describe the introduction and review of the work.

Response

Thanks for your comment. We have reorganized and enriched our introduction. In this part, we first briefly describe the functions of ncRNA in plants and its proportion in the genome (line 28-36). Then we introduce the category of ncRNA in plants and the types of ncRNA that have not been described yet (line 37-48). Finally, we summarize the four main contents (biogenesis, functions, interactions, and resources) to help researchers better understand the world of ncRNAs at multiple levels (line 49-68). Notably, we also make a detailed statement of these four aspects at the beginning of each part in this review.

In Conclusion, the authors should add the significance of this research, and its potential practical application. Conclusion should be separate.

Response

Thanks for your suggestion. As this review does not have a separate conclusion section, we are a little bit confused by the reviewer's comment. We made the following modifications to the Discussion section. First, we summarize this review again (line 393-402), and supplement the deficiencies of circRNAs in plant research (line 403-410). Furthermore, we make some prospects for the research of ncRNA at the single-cell level (line 411-419). Additionally, we have added a separate Conclusion section to describe potential practical applications of this review that will help the scientific community to better understand the complex ncRNA world (line 421-427).

The MS English needs to be improved. The article's English must be carefully checked for grammatical errors.

Response

Thank you so much for your careful check. We have carefully revised our manuscript, hope this version to meet your requirement.

Reviewer 2 Report

The manuscript submitted by the authors discusses and reviews the updated information about Non-coding RNAs (ncRNAs). The manuscript is well written and the figures, tables are quite informative. I have some minor comments, I  would suggest adding a bit more information about circRNAs when authors talk about this section and the discussion part to be a bit elaborated. Besides, there are a few minor spelling mistakes in the figure and text.

Author Response

Response to Reviewer 2 Comments

The manuscript submitted by the authors discusses and reviews the updated information about Non-coding RNAs (ncRNAs). The manuscript is well written and the figures, tables are quite informative. I have some minor comments, I would suggest adding a bit more information about circRNAs when authors talk about this section and the discussion part to be a bit elaborated.

Response

We gratefully appreciate your valuable comments and made corresponding modifications. In the second part, we elaborate on three ways of circRNAs biogenesis, and describe the functions of circRNAs in plant development and stress response combined with previous studies at line 206-218. In discussion, we summarize the main contents of this review in detail, and supplement the deficiencies of circRNAs in plant research at line 403-410. Furthermore, we make some prospects for the research of ncRNA at the single-cell level at line 411-419. Finally, we summarize the work and objectives of this review again in conclusion at line 421-427.

Besides, there are a few minor spelling mistakes in the figure and text.

Response

Thank you so much for your careful check. We have carefully checked the manuscript and corrected such mistakes. Hope this version meet your requirement.

Round 2

Reviewer 1 Report

Requested corrections were completed.